# Molecular diagnosis of visceral leishmaniasis: a French study comparing a reference PCR method targeting kinetoplast DNA and a commercial kit targeting ribosomal DNA

Grégoire Pasquier,[1] Quentin Andreotti,[1] Christophe Ravel,[1] Yvon Sterkers[1]

**ABSTRACT** Visceral leishmaniasis is a life-threatening disease caused by the *Leishmania donovani* species complex. For direct diagnosis, molecular diagnosis on blood and bone marrow is an increasingly used technique. For *Leishmania*, most PCR assays are laboratory-developed, but marketed PCR assays are now available and should be evaluated independently of manufacturers. The "quanty Leishmaniae, Clonit" kit was compared to a laboratory-developed method widely used in France and considered here as a reference method. Performances were evaluated on serial dilution assays, on 5 external quality controls and on 35 clinical samples. The reference method performed better than the Clonit kit with higher "performance scores": 20 of 28 (71%) and 4 of 12 (33%) vs 17 of 28 (61%) and 1 of 12 (8%) for the reference method and the Clonit kit, respectively. On clinical samples, six false negative results out of 27 positive samples (22%) were observed with the quanty Leishmaniae, Clonit method. These results are most likely due to the difference in the number of repeats of the PCR targets. The quanty Leishmaniae, Clonit, like most of marketed methods, targets the ribosomal DNA that has a lower number of copies than the kinetoplast DNA targeted by the reference PCR. This study confirms that the choice of target is crucial and should be taken into account in the development of new highly sensitive PCR methods.

**IMPORTANCE** PCR revolutionized the direct diagnosis of infectious diseases, especially protozooses, where the infectious load is usually low. Commercial PCR methods are available and offer many advantages, including convenience and batch tracking as part of a quality system. For most parameters, the performance of commercial methods is at least as good as that of finely optimized methods developed in expert laboratories. This comparison work has not been done for the molecular diagnosis of visceral leishmaniasis. *Leishmania* sp. has a unique organelle, the kinetoplast, which corresponds to the mitochondrial DNA. It is organized into a large number of minicircles, which has made it a target for the development of diagnostic PCR. The quanty Leishmaniae, Clonit kit targeting ribosomal DNA was compared to a widely used laboratory-developed method based on kinetoplast DNA. This reference method gave significantly better results, probably due to the difference in the number of repeats of the PCR targets.

**KEYWORDS** leishmaniasis, qPCR, comparison method, quantification, real-time PCR, National Reference Center for Leishmaniasis

Leishmaniasis are a group of diseases ranging from cutaneous leishmaniasis (CL) and muco-cutaneous leishmaniasis (MCL) to visceral leishmaniasis (VL) forms caused by various species of a flagellated protozoa of the genus *Leishmania* and transmitted by phlebotomine sandflies (1). Among the different clinical forms, VL is less incident but much more severe than CL and MCL, being most often fatal without appropriate

Address correspondence to Yvon Sterkers, yvon.sterkers@umontpellier.fr.

The authors declare no conflict of interest.

treatment. According to the World health Organization (WHO), leishmaniasis is a neglected tropical disease and remains a major health issue in the Americas, North Africa, West and Southeast (Leishmaniasis Key facts, https://www.who.int/news-room/fact-sheets/detail/leishmaniasis, last visited 14 August 2023). According to the WHO Leishmaniasis Control Team, there were between 200,000 and 400,000 estimated cases of VL worldwide in 2012 (2). In 2017, updated data showed a decrease with between 50,000 and 90,000 estimated VL cases (1). VL is due to *Leishmania infantum* in the Mediterranean basin, including South of France, and in South America, and to *Leishmania donovani* in the Indian subcontinent and in East Africa. These two species are closely phylogenetically related and belong to the *L. donovani* species complex (3); they are sensitive to the same treatments (1). Thus, unlike CL, precise species identification is not necessary for clinical and therapeutic management of VL. Autochthonous VL cases in metropolitan France mainly affect immunosuppressed patients (AIDS and solid organ transplanted patients and patients undergoing immunosuppressive therapy) and infants under 6 years old (4). Microbiological diagnosis of VL can be made by direct microscopic examination, serological tests, cultures, and PCR techniques (1). de Ruiter et al. in a meta-analysis (5) reported a pooled sensibility of 93.1% and a specificity of 95.6% in blood. Among these molecular technics, quantitative PCR (qPCR) (or real-time PCR) gave the best results in terms of sensitivity and specificity (6). In addition, qPCR could be used to monitor parasite load for follow-up and is informative for assessing disease prognosis (7). Several commercial DNA-based diagnosis kits are now available for the diagnosis of VL from peripheral blood samples (8), and most of them target the 18S ribosomal RNA sequences. In practice, many parameters can have an impact on PCR assay performances, such as the DNA extraction method, the copy number of the target gene, and the design of the primers (8). The need for best sensitivity is more acute in parasitology and mycology than in other fields of microbiology because the microbial loads are most often lower. The 18S rRNA gene is well conserved and multicopy, making it a widely used target for the diagnosis of many protozoa and fungi. In *Leishmania* spp., the rRNA genes are organized on chromosome 27 in a large tandem array (9). The copy number of the 18S rDNA gene is estimated to be about 20–40 (10). Like other kinetoplastids, *Leishmania* spp. have a kinetoplast which is a specialized region of their unique mitochondria (11). This organelle contains circular DNAs (kDNA), about 50 maxicircles (20–40 kb in size) coding for the respiratory chain and 5,000 and 10,000 of minicircles (0.5–1 kb in size) coding for RNAs involved in the regulation of maxicircle gene expression (11). The high number of minicircles makes them a prime target for molecular diagnosis, even though these molecules are difficult to sequence and genetically heterogeneous. There is no consensus on the choice of target for VL molecular diagnosis (12). Two ancient studies reporting data obtained by conventional, non-quantitative, PCR in dogs and in MCL showed that the sensitivity of the kDNA targeting method could be superior to the rDNA targeting method (13, 14). The purpose of this study was to compare and evaluate the performances of the "quanty Leishmaniae, Clonit" method targeting the rDNA and a finely optimized and highly sensitive reference qPCR method developed by Mary et al. targeting minicircles in kDNA (15). The comparative study presented here was divided into three parts. The first part used DNA standards in a one-tenth serial dilution assay to determine the PCR efficiencies of the two methods. The second part used external quality controls (EQCs) to assess the performances of both qPCR technics on inter-laboratory comparison samples. The last part used patient samples to evaluate the performance and the agreement of the quanty Leishmaniae, Clonit, and the reference methods on clinical samples.

## MATERIALS AND METHODS

### Samples

Three kinds of samples were included in this study: standards, EQC samples, and clinical blood samples. First, *L. infantum* (MHOM/MA/67/ITMAP263, clone S9F1) *in vitro* grown as promastigote forms in HOMEM culture medium (16) and harvested at a concentration of $10^6$*Leishmania*/mL was used to make standards. Standards were obtained from dilutions of extracted *Leishmania* DNA. Second, five EQC samples were included. They were provided by the Centre Toulousain pour le Contrôle de qualité en Biologie clinique (CTCB) and corresponded to all EQC samples received by the routine laboratory between 2020 and 2022 in frame with the ISO 15189 accreditation maintenance. Four samples were positive and one was negative. Third, 35 clinical blood samples were included in this study. They corresponded to the 27 positive samples corresponding to the 23 consecutive patients analyzed for VL at Montpellier University Hospital between April 2019 and March 2022 and 8 control samples/patients from the same period of time (Fig. 1). Criteria for a definite diagnosis of VL were (i) clinical signs and symptoms consistent with VL, (ii) positive *Leishmania* serology or qPCR on other specimens and (ii) a clinically retained diagnosis of VL and appropriate treatment of VL (see Table S1). The eight control patients were clinical specimens that were negative in routine PCR testing of positive patients and in whom the diagnosis of VL had been excluded (see Table S1).

### DNA extraction methods

Standards were extracted using the QIAamp DNA mini kit (cat no 51306; Qiagen, Courtaboeuf, France) with 400 µL of culture for input and 200 µL for elution following manufacturer instructions. DNA from clinical samples was extracted using the QIAamp DNA mini kit (cat no 51306, Qiagen) from the buffy coat before January 2021 and then, from whole blood, in a MagNA pure compact using the MagNA Pure Compact Nucleic Acid Isolation I kit (cat no 03730964001; Roche, Meylan, France) using the same input and elution volumes as the manual extraction method.

### PCR methods

The reference qPCR method is an in-house qPCR method developed 20 years ago by Mary et al. and targets a conserved region of the kDNA (15). This method is used by 10 of 11 French laboratories which routinely perform VL molecular diagnosis and have answered a survey on laboratory practices realized by the French National Reference Center for Leishmaniasis in February 2023. The product length is ~120 bp, and revelation of the PCR product is based on hydrolysis probe (TaqMan). Primer pairs and probe sequences are the following: RV1-CM-F 5′CTTTTCTGGTCCTCCGGGTAGG, RV2-CM-R 5′CCA CCCGGCCCTATTTTACACCAA and minicircle_prob_1 5′ [YAKYE]-TTTTCGCAGAACGCCCCTA CCCGC-[ECLIP] (Eurofins Genomics, Ebersberg, Deutschland). Reference qPCR runs were performed according to the good practice rules: in duplicate, including a positive control, a negative control and an internal control. The positive control is made of a plasmid which contains a modified target sequence. It is used at a defined concentration corresponding to a crossing point (Cp) value at 29.7 ± 0.4. To validate a run, the Cp value of the positive control was added to a Lewey-Jenning plot and Westgard rules were applied. To detect the presence of PCR inhibitors, an internal control consisting of the positive control is added to the DNA patient sample in a third reaction well. The presence of inhibitors is excluded if the difference between the Cp values of the positive control and the internal control is less than three. In addition, qPCR targeting the albumin gene was performed in a fourth reaction well as a cellularity control. Primer pairs and probes are the following: Albu-F 5′ GCTGTCATCTCTTGTGGGCTGT, Albu-R 5′ ACTCATGGGAGCTGC TGGTTC and Albu-Probe 5′ [FAM]-GGAGAGATTTGTGTGGGCATGACA-[BHQ1] (17). The PCR sample uptake was 2 µL in the final reaction volume of 20 µL. Runs were performed on a LightCycler 480 (Roche). The PCR amplification process included 45 cycles for 10 s at 95°C, 30 s at 60°C. We have accredited this method through the ISO 15189 international

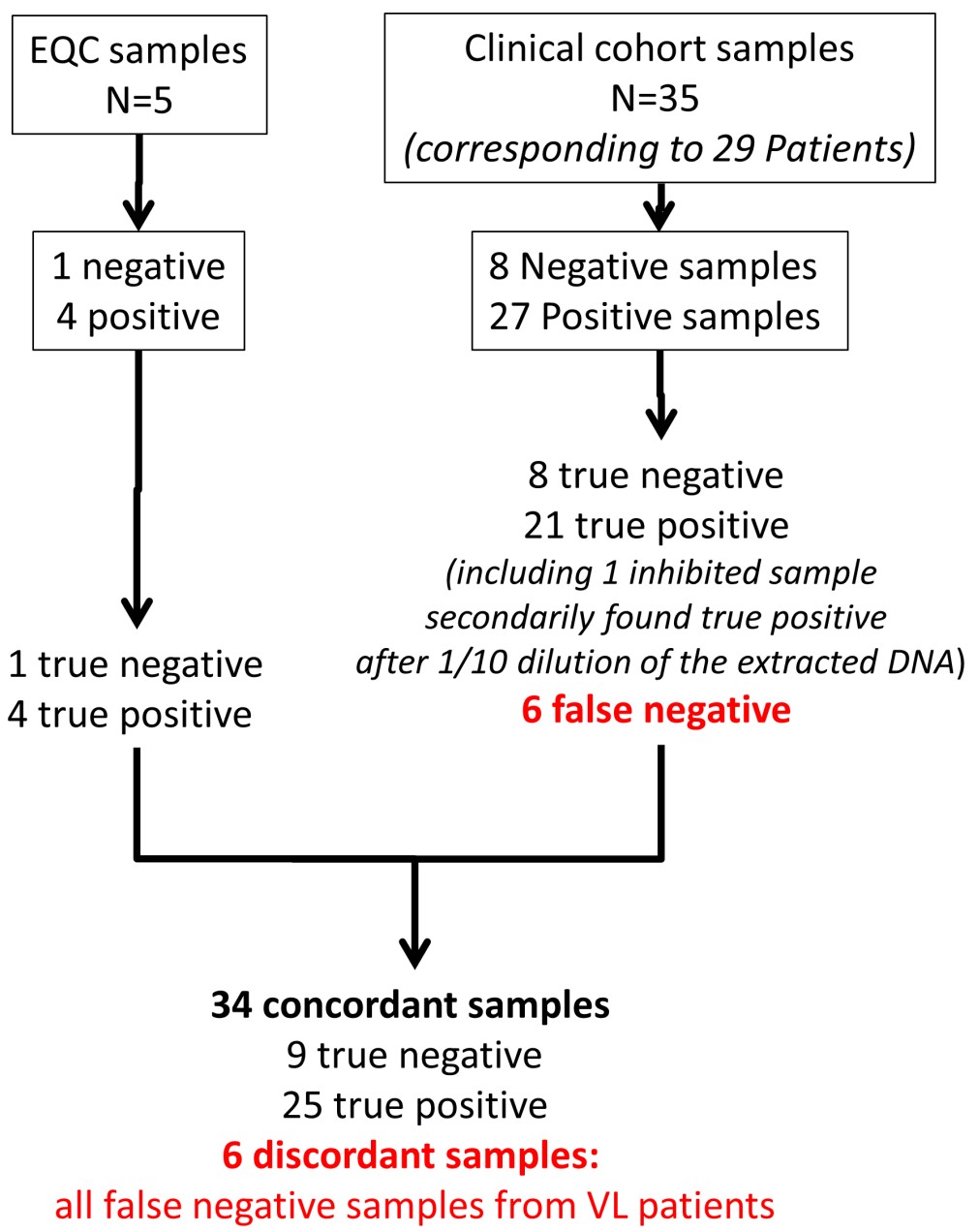

FIG 1 Flowchart of the 40 analyzed samples. The comparison study was based on five external quality control (EQC) samples, on a clinical cohort of 15 patients and on 4 additional clinical samples which corresponded to VL patients with low parasitemia.

standard in 2017. The quanty Leishmaniae, Clonit (Ref RT-63q; Launch Diagnostics, Paris, France) qPCR method is also based on hydrolysis probe (TaqMan) but targets the 18S ribosomal RNA sequences. The qPCR runs were performed in duplicate on a LightCycler 480 (Roche) according to the manufacturer procedure. The quanty Leishmaniae, Clonit qPCR kit is a CE-marked diagnostic kit according to the European *in vitro* diagnostic directive 98/79/CE. The kit contains reagents enough to perform 48 amplification tests. The kit contains seven easily identifiable tubes with colored caps: R1 tubes correspond-ing to the amplification mixture, R2 tubes containing the "Leishmaniae probes," a negative control (R7 tube) and for quantification purposes, R3–R6 tubes containing cloned 18S rDNA gene at concentrations ranging from $10^5$ to $10^2$ copies/µL. According to the manufacturer, to validate the analysis from the extraction process to the detection

**TABLE 1** Comparison of Cp of the two qPCR methods obtained in the one-tenth serial dilution assays

| | Conc. (Leish./mL)[a] | $10^6$ | $10^5$ | $10^4$ | $10^3$ | 100 | 10 | 1 | 0.1 | 0.01 | Total score/28 | Most stringent score/12 |
|---|---|---|---|---|---|---|---|---|---|---|---|---|
| Reference | Qualititative result | 2/2 | 2/2 | 2/2 | 2/2 | 4/4 | 4/4 | 3/4 | 1/4 | 0/4 | 20 (71%) | 4/12 (33%) |
| | Cp value (mean ± SD)[b] | 17.1 | 20.3 | 23.5 | 27.1 | 30.6 ± 0.47 | 34.9 ± 0.52 | 38.8 ± 1.44 | 40.0 | Neg | | |
| "Quanty Leishmaniae, Clonit" | Qualititative result | 2/2 | 2/2 | 2/2 | 2/2 | 4/4 | 4/4 | 1/4 | 0/4 | 0/4 | 17 (61%) | 1/12 (8%) |
| | Cp value (mean ± SD)[b] | 22.4 | 26.4 | 29.9 | 33.4 | 36.5 ± 0.59 | 39.9 ± 0.27 | 40.0 | | | | |

[a]Conc., concentration, Leish./mL, *Leishmania* per milliliter.
[b]SDs were calculated when applicable, i.e., when ≥3 Cp values were obtained. "Total score/28" corresponds to the total number of positive wells out of the total number of wells. "Most stringent score/12" corresponds to the number of positive wells in the three lowest concentrations assayed from 1.0 to 0.01 *Leishmania*/mL. SD, standard deviation.

step, the Cp values for the internal control (beta-globin) must not be greater than 28 or undetermined. The PCR sample uptake was 5 µL in a final reaction volume of 25 µL. The PCR amplification process included 45 cycles with 15 s at 95°C and 60 s at 60°C. For both methods, the absolute quantification/second derivative maxima method (Light-Cycler 480 software version 1.5.1.62; Roche) was used to calculate the Cp values.

## Analysis of performances and statistical analysis

From a standard DNA sample corresponding to $10^6$ *Leishmania*/mL, eight one-tenth dilutions were done. The four highest concentrations were assessed in duplicate and the lowest from 100.0 to 0.01 *Leishmania*/mL were assessed in tetraplicate. Therefore, a total number of 28 wells were performed (see Table 1). The qPCR performances were assessed by determining the range of linearity of the standard curve, and the PCR efficiencies were calculated using the slope of the trend line of the standard curve [efficiency (EFF) = $10^{(-1/slope)} - 1$]. In order to facilitate the analysis of the performance of the methods, two scores were elaborated as described elsewhere (18). The "total score/28" is the sum of positive wells out of the total number of wells, and the "most stringent score/12" focused on the data obtained in the three lowest concentrations. For qualitative results, the agreement was evaluated with the Cohen's kappa coefficient. For quantitative results, agreement and correlation between the two qPCR technics were assessed thanks to a Bland-Altman plot. Cp pairwise comparison was tested with a pairwise Wilcoxon test and illustrated with a boxplot thanks to the R version 4.2 software.

## RESULTS

### Analytical sensitivity of the reference method compared to the Clonit method

Nine one-tenth DNA dilutions corresponding to *Leishmania* concentrations ranging from $10^6$ to 0.01 *Leishmania*/mL were analyzed by both methods, which allowed us to obtain a standard curve. PCR efficiencies were determined based on the slope of the standard curve (EFF = $10^{(-1/slope)} - 1$) and were calculated at 92% for the reference qPCR and 95% for the Clonit kit. With $R^2$ >0.995 for both tests (0.998 and 0.999 for reference and quanty Leishmaniae, Clonit qPCR tests, respectively), zones of linearity were similar, spanning on 6 log from 10 to $10^6$ *Leishmania*/mL (Fig. 2). The total score/28 [see Materials and Methods and reference (17)] was 20/28 (71%) for the reference method and 17/28 (61%) for the Clonit kit, and the most stringent scores/12 were 4/12 (33%) and 1/12 (8%), respectively (Table 1).

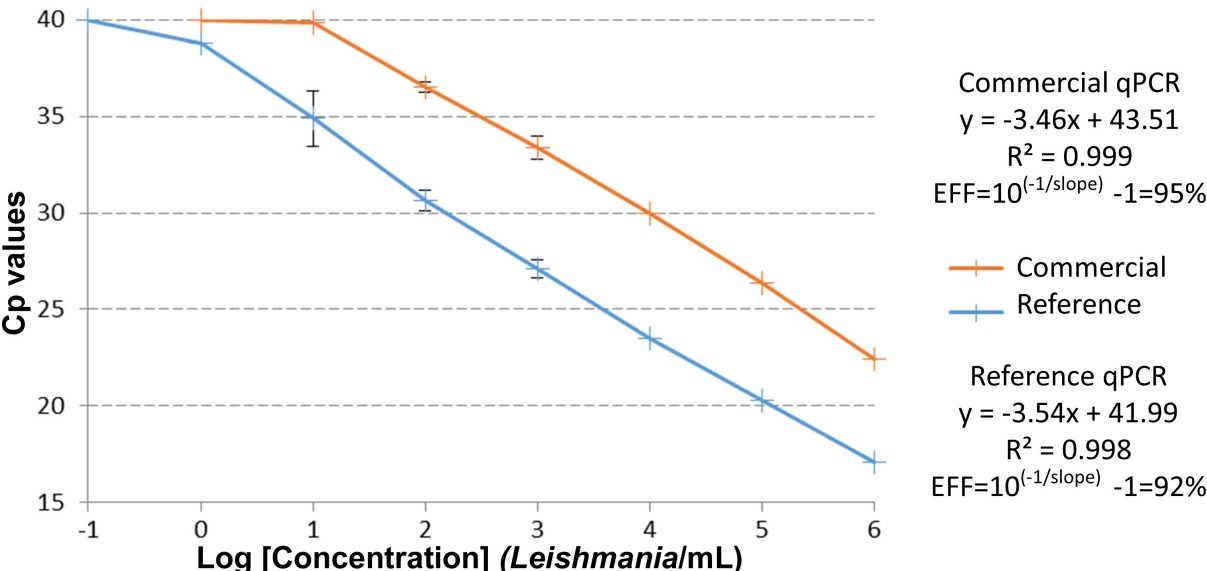

**FIG 2** Linearity zone and PCR efficiency (EFF) estimated from the standard curves obtained in the one-tenth serial dilution assays.

## Concordant results with external quality control samples

Five EQC samples from the CTCB quality control provider were tested in duplicate with both qPCR methods. All results were concordant: the four positive EQC samples were correctly assessed as positive and the negative EQC samples as negative by both methods (Table S1).

## Sensitivity of the reference method compared to the Clonit kit on clinical samples

Thirty-five blood DNA extracts were analyzed by both methods. The diagnosis of VL was retained for 27 patients and excluded for eight patients (see Materials and Methods section, Fig. 1; Table S1). Regarding the qualitative results, the confusion matrix (Table 2) showed a Cohen's kappa agreement coefficient of 0.62 with 21 patient samples positive in both methods, 8 negative in both cases, and 6 discordant samples. These samples must be considered as false negative results of the Clonit kit because the diagnosis of VL was affirmed on (i) a compatible clinical history of VL, (ii) a positive *Leishmania* serology, and (iii) several positive qPCR samples on different matrixes: blood, bone marrow, and lymph nodes (see Table S1). In routine practice, these blood samples were found positive with a Cp = 38.7, which corresponds to a very low parasite burden (estimated close or below the limit of quantification of the method at about 1 *Leishmania*/mL of blood). An internal control (IC) to check for the quality of DNA extraction and to detect PCR inhibition is included in both methods. In seven samples, the Cp value of the IC was above the threshold of 28 (Table S1). Sample 12, patient 6, had the reference PCR inconstantly positive/partially inhibited when tested pure in tetraplicate (negative twice, 35.6 and 40.0) and positive when diluted one-tenth (27.9, 27.3, and 27.5). With the Clonit kit, it was totally inhibited with a negative internal control when tested pure in tetraplicate. The negativity of this control led to a second analysis after a one-tenth dilution of the extracted DNA; late Cp values where obtained after one-tenth dilution (40.0 and 38.7). These late Cp values probably witnessed persistence of a partial inhibition of the PCR reaction with the Clonit kit. However, the presence of an IC in the kit and the one-tenth dilution allowed this sample to be positive and not falsely negative.

**TABLE 2** Confusion matrix between the two qPCR methods[a]

|  |  | "Quanty Leishmaniae, Clonit" | |
|  |  | Positive | Negative |
| --- | --- | --- | --- |
| Reference | Positive | 21 | 6 |
|  | Negative | 0 | 8 |

[a]Confusion matrix reporting the qualitative results obtained on the 35 results obtained from the clinical cohort. Cohen's kappa agreement coefficient = 0.62 (substantial agreement).

## Agreement between the quantitative results obtained with both methods

In order to test the agreement between the quantitative results (Cp values) obtained with both methods, we performed a Bland-Altman plot with the 25 EQC and clinical samples found concordantly positive using both methods. Both methods present a good agreement, with only two samples outside the ±1.96 standard deviation range in a Bland-Altman plot (Fig. 3A). We observed a difference of Cp values at 5.8 ± 0.6 in the linear zone of the standard curves obtained in the serial dilution tests and at 6.0 ± 2.5 with the 21 concordant positive clinical samples. We performed a pairwise comparison of the Cp values obtained with both qPCR methods; the Cp values obtained with the Clonit kit were significantly later ($P < 0.0001$) than those obtained by the reference method (Fig. 3B; Table S1).

## DISCUSSION

The parasitology-mycology laboratory of the Montpellier University Hospital hosts the French National Reference Center for Leishmaniasis. Approximately 1,000 clinical samples are referred to and analyzed in the laboratory for VL and CL diagnoses per year. The rationale for this study is that more and more commercial kits for the molecular diagnosis of VL are available. Shifting to a marketed PCR kit makes sense in terms of usability, quality assessment, and standardization. However, it should not be at the expense of performances which must be similar to previous laboratory-developed reference methods. To our knowledge, no studies analyzing the performances of commercial assays have been evaluated independently of manufacturers and published for the molecular diagnosis of VL. Here, we found that both methods were finely optimized and had good PCR efficiencies: 92% for the reference qPCR and 95% for the quanty

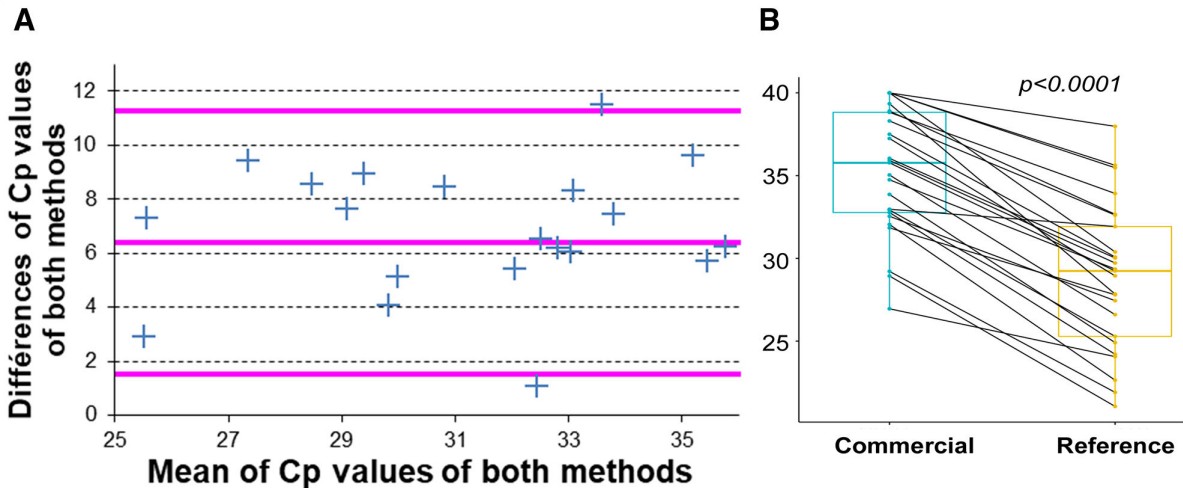

**FIG 3** Agreement and pairwise comparison of crossing point (Cp) values the two qPCR methods (A). Bland-Altman plot representing the concordance between the Cp values obtained using the reference and the "quanty Leishmaniae, Clonit" qPCR for molecular detection of *Leishmania infantum*. (B) Boxplot representation of the pairwise comparison of the Cp values obtained with the commercial (left) and reference (right) qPCR methods on the 25 samples (EQCs and clinical samples) concordantly found positive with both methods. The Cp values obtained with the commercial method were later than those obtained by the reference method. We used a pairwise Wilcoxon test to show that the difference is highly significantly significant ($P < 0.0001$).

Leishmaniae, Clonit qPCR. The zone of linearity for both methods was similar and large, spanning on six logs from 10 to $10^6$ *Leishmania*/mL (Fig. 2). PCR efficiency and zone of linearity determined here are in agreement with those given by the manufacturer in the technical data sheet. However, the reference method performed better than the Clonit kit. First, the performance scores [see Materials and Methods and reference (18)] were slightly better for the reference method as compared to the Clonit kit (Table 1). Second, the results obtained on clinical samples were also better with the reference method as compared to the Clonit kit. In this study, there were six false-negative results in the cohort of 35 samples. This lower sensitivity is a major drawback because the cohort corresponds to our routine diagnostic activity and therefore covers the range of parasite loads seen in daily practice. Over more than 10 years, it can be estimated that about a quarter of the positive samples are positive with a load of less than 10 *Leishmania*/mL of blood (National Reference Center for Leishmaniasis, unpublished data). To our opinion, this lower sensitivity is mainly due to the choice of the DNA target and in particular to the difference in the number of targets. In the linear zone of the standard curves obtained in the serial dilution tests, we measured a difference of Cp values of $5.8 \pm 0.6$ between the two methods (Fig. 3B). Using 5.8 for the Cp shift and 100% for the PCR efficiency of both techniques, we found it can be estimated that there is $2^{5.8} = 58$ more kinetoplast targets compared to rDNA targets. Taking into account the DNA input volumes of 2 and 5 µL for the reference and quanty Leishmaniae, Clonit qPCRs, respectively, and multiplying 58 by 2.5 gives 145. This value is in the order of magnitude of the 125–500 times more copies of the kDNA target compared to the rDNA target obtained from the figures quoted above. This value is in the low range of the estimate, possibly due to factors other than the copy number of the target, such as catenation of the minicircles forming the kDNA (11), which in turn may reduce the extraction yield and prevent the correct amplification of some minicircles during PCR. The quanty Leishmaniae, Clonit kit includes controls (see Materials and Methods) to perform a standard curve and DNA quantification. We do not present data comparing the quantification data obtained by the two systems because the different nature of the DNA targets as well as the use of different units for the quantification curves, copies per microliter vs *Leishmania* per milliliter severely hinder this comparison in our opinion.

To conclude, the Clonit kit is less performant to detect low parasitemia than the laboratory-developed reference qPCR. This method is therefore not sensitive enough and can lead to false-negative results when parasite load is low. This is a major concern because, in routine practice, a quarter of positive samples have a parasite load of less than 10 *Leishmania*/mL of blood. This is probably due to the fact that the kit targets rDNA, while the reference method targets kDNA. Further development of commercial kits should take this into account.

## ACKNOWLEDGMENTS

We are grateful to Sylvie Douzou, France Joullié, and Martine Brun for their technical help. We warmly thank the participants in the survey on laboratory practices carried out by the National Reference Center for Leishmaniasis and distributed by the learned society Association Française des Enseignants & Praticiens Hospitaliers titulaires de Parasitologie & Mycologie Médicale (French Association of Lecturers & Hospital Practitioners of Parasitology & Medical Mycology) (ANOFEL).

Extramural funding was received from "Santé Publique France" through the French National Reference Center for Leishmaniasis and from the RSI Assurance Maladie Professions Libérales-Provinces for funding the LightCycler 480 Real-time Thermocycler (Roche). The funders had no role in study design, data collection and interpretation, or the decision to submit the work for publication.

## AUTHOR AFFILIATION

[1]Department of Parasitology-Mycology, National Reference Center for Leishmaniasis, University of Montpellier, CNRS, IRD, University Hospital Center (CHU) of Montpellier, Montpellier, France

## AUTHOR ORCIDs

Yvon Sterkers  http://orcid.org/0000-0002-5623-5664

## AUTHOR CONTRIBUTIONS

Grégoire Pasquier, Data curation, Formal analysis, Funding acquisition, Investigation, Methodology, Writing – original draft, Writing – review and editing | Quentin Andreotti, Formal analysis, Investigation, Writing – original draft | Christophe Ravel, Formal analysis, Funding acquisition, Investigation, Resources, Validation, Writing – original draft, Writing – review and editing | Yvon Sterkers, Conceptualization, Data curation, Formal analysis, Funding acquisition, Methodology, Project administration, Resources, Supervision, Validation, Writing – review and editing

## ETHICS APPROVAL

This study was realized with the approval of the local medical ethics committee of the University Hospital Center of Montpellier, France, in line with the revised Helsinki Declaration.

## ADDITIONAL FILES

The following material is available online.

### Supplemental Material

**Table S1 (Spectrum02154-23_S0001.xlsx).** Raw data.

### Open Peer Review

**PEER REVIEW HISTORY (review-history.pdf).** An accounting of the reviewer comments and feedback.

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
