## [Reviewer comments · Microbiology Spectrum]

Microbiology Spectrum

Molecular diagnosis of visceral leishmaniasis a French study comparing a reference PCR method targeting kinetoplast DNA and a commercial kit targeting ribosomal DNA.

Grégoire Pasquier, Quentin Andreotti, Christophe Ravel, and Yvon Sterkers

Corresponding Author(s): Yvon Sterkers, Universite de Montpellier

Review Timeline:

Submission Date:	May 22, 2023
Editorial Decision:	June 19, 2023
Revision Received:	August 18, 2023
Accepted:	August 23, 2023

Editor: Denis Sereno

Reviewer(s): The reviewers have opted to remain anonymous.

Transaction Report:

DOI: <https://doi.org/10.1128/spectrum.02154-23>

June 19, 2023

Dr. Yvon Sterkers
Universite de Montpellier
Montpellier cedex 5
France

Re: Spectrum02154-23 (Molecular diagnosis of visceral leishmaniasis, a reference PCR method targeting kinetoplast DNA performs better than a commercial kit targeting ribosomal DNA.)

Dear Dr. Yvon Sterkers:

Link Not Available

Sincerely,

Denis Sereno

Journals Department
Reviewer comments:

Reviewer #1 (Comments for the Author):

The article entitled: "Molecular diagnosis of visceral leishmaniasis, a reference PCR method targeting kinetoplast DNA performs better than a commercial kit targeting ribosomal DNA" describes a comparison study between two real-time PCR techniques, in-house vs commercial qPCR. The comparison has been performed in both qualitative and quantitative terms. The work is correct, although the results include phrases that would correspond to the discussion section. An important limitation of this study, which it was not mentioned in the Discussion section, is the small number of samples of the study. Indeed, the theoretical difference between the two qPCRs is the number of copies of both targets. That is, considering that the number of copies of the conserved sequence of kDNA is 10,000 and that of the 18S ribosomal gene is 200 in Old World Leishmanias (Table 3 of the reference 8 of this study, Gow et al 2022; Van Eys et al, 1992, Van Eys et al, 1989), it is expected that the

analytical sensitivity of kDNA-based qPCR is higher than that of ribosomal gene-based qPCR. The authors do not mention other studies carried out in Latin America (León et al, 2017), where kDNA-based qPCR presents specificity drawbacks, which is to be expected, since the higher is the sensitivity, the lower is the specificity. In clinical diagnosis it is important to maintain a good balance between both parameters. Samples with low parasitaemias are a challenge even for extremely sensitive techniques.

COMMENTS (according to word document)

Title: It could be convenient to point out that the study was performed in France. The number of samples that was included in the study is low to support the title, from my point of view. Title could be: "Molecular diagnosis of visceral leishmaniasis in France: comparison between of a reference PCR method targeting kinetoplast DNA and a commercial kit targeting ribosomal DNA."

Line 15: "For direct diagnosis, quantitative PCR in blood is currently the most commonly used technique."
Please review this statement, according to the literature, PCR in blood is not the most common for the diagnosis of VL.

Line 19: Add: in France. The reference method, the authors stated, is not the reference method in everywhere.

Line 29: "Leishmaniasis are protean diseases..."
Please clarify this sentence.

Line 58: The copy number of the 18S rDNA gene is estimated to be about 20-40 (10)."
According to the reference 8, the number of copies is around 200.

Line 151-162: "The rationale for this study is that more and more commercial kits for the molecular diagnosis of VL are available. The use of commercial kits should be encouraged, especially in the context of good laboratory practice and diagnostic accreditation. Nevertheless, it is fundamental that the performance of commercial kits be analyzed by experts independent of the manufacturer. This study was performed in the Parasitology-Myology laboratory of the Montpellier University Hospital which hosts the French National Reference Center for leishmaniasis. Approximately 1,000 clinical samples are referred to and analyzed in the laboratory for VL and CL diagnosis per year. The study was divided into three parts. The first part used DNA standards in a one-tenth serial dilution assay to compare PCR efficiencies and to approach the sensitivity limits of the two methods. The second part used external quality controls (EQCs) to assess the performances of both qPCR technics on inter-laboratory comparison samples. The last part used patient samples to evaluate the performance and the agreement of the "quany Leishmaniae, Clonit" and the reference methods on clinical samples."
This paragraph could go on Discussion section.

Line 163: Delete: "Better". The reader should be able to evaluate the data and to reach the similar conclusion.

Add Probit analysis to compare the limit of detection between both tests.

Line 179: Delete: "Better"

Consider the following references:

León CM, Muñoz M, Hernández C, Ayala MS, Flórez C, Teherán A, Cubides JR, Ramírez JD. 2017. Analytical Performance of Four Polymerase Chain Reaction (PCR) and Real Time PCR (qPCR) Assays for the Detection of Six Leishmania Species DNA in Colombia. *Front Microbiol* 8:1907. <https://doi.org/10.3389/fmicb.2017.01907>

van Eys GJ, Schoone GJ, Ligthart GS, Alvar J, Evans DA, Terpstra WJ. Identification of 'Old World' Leishmania by DNA recombinant probes. *Mol Biochem Parasitol.* 1989 Apr;34(1):53-62. doi: 10.1016/0166-6851(89)90019-4

van Eys GJ, Schoone GJ, Kroon NC, Ebeling SB. 1992. Sequence analysis of small subunit ribosomal RNA genes and its use for detection and identification of Leishmania parasites. *Mol Biochem Parasitol* 51:133-142. [https://doi.org/10.1016/0166-6851\(92\)90208-2](https://doi.org/10.1016/0166-6851(92)90208-2)

Reviewer #2 (Comments for the Author):

Major comments

The study presented by Pasquier and his colleagues aims to compare a commercial visceral leishmaniasis diagnostic kit with a non-marketed but widely used reference method. Results show that the commercial kit does not reach the performance of the reference method, giving rise to several false negative results. The reviewer partially subscribes to the conclusions of the authors, nevertheless, he finds that their demonstration deserves to be better supported. This will be the meaning of the comments presented below.

The main problem is the low number of samples tested in the successive phases of the study. The study relies on 5 external quality control and 19 clinical samples. They have no discordant results on the 5 quality controls (4 positive and 1 negative) and only 1 discordant result on the 15 samples tested from their routine practice (sample 6, patient 1). Sample 12 from patient 6 is not really a discordant one since the authors identified the presence of an inhibitor in this sample and could solve the problem after dilution.

Finally, the main differences between the results obtained with the two methods is highlighted on a complementary set of four samples. Notably, these samples were selected because they had weak positive results with the reference method. Pasquier et al further show that the commercial technique failed to detect Leishmania DNA in three of these four samples.

From a methodological point of view, this way to select samples induces a bias. These samples are neither representative of the positive samples (since they are selected to be at the detection limit of the reference method), nor representative of all the samples tested in routine practice (which are mostly negative samples).

As a result, it is impossible to perform a sensitivity and specificity calculation. Furthermore, there is no estimate of the performance of the reference method in terms of sensitivity and specificity. Here, the specificity seems to be 100%, but the number of negative patients tested (5) is insufficient. As for the sensitivity, it cannot be calculated since the positivity of the PCR was one of the criteria for including four out of the 14 positive patients tested.

Nevertheless, the hypothesis formulated by Pasquier and his co-authors seems acceptable to me and I recommend continuing the study by including a greater number of routine samples in the comparison. Doing so they will probably obtain significant differences when estimating the performance of both methods and provide more convincing results.

Other comments

The abstract does not present any numerical value, only sentences indicating that the reference method gives better results.

The results section begins with two paragraphs which relate either to the description of the objectives of the study and its rationale (first paragraph), or to the description of the method and design of the study. It would be necessary to integrate them into the corresponding sections of the article.

Staff Comments:

Preparing Revision Guidelines

Please return the manuscript within 60 days; if you cannot complete the modification within this time period, please contact me. If you do not wish to modify the manuscript and prefer to submit it to another journal, please notify me of your decision immediately so that the manuscript may be formally withdrawn from consideration by Microbiology Spectrum.

Re: Spectrum02154-23 (Molecular diagnosis of visceral leishmaniasis, a reference PCR method targeting kinetoplast DNA performs better than a commercial kit targeting ribosomal DNA.)

Dear Editor

We answered to all the points raised by reviewer #1 and #2. In particular we increased the number of included samples, which was 20 in the first version of the manuscript and raised to 40 samples (corresponding to 29 patients) in this version. Please find below the point-by-point responses.

Point-by-point responses to the issues raised by the reviewers

Reviewer #1 (Comments for the Author):

The article entitled: "Molecular diagnosis of visceral leishmaniasis, a reference PCR method targeting kinetoplast DNA performs better than a commercial kit targeting ribosomal DNA" describes a comparison study between two real-time PCR techniques, in-house vs commercial qPCR. The comparison has been performed in both qualitative and quantitative terms.

The work is correct, although the results include phrases that would correspond to the discussion section.

We have removed the word "better" from two sub-section headings and transferred the paragraph concerning the difference in Cp values between the methods to the discussion section.

COMMENTS (according to word document)

Title: It could be convenient to point out that the study was performed in France. The number of samples that was included in the study is low to support the title, from my point of view. Title could be: "Molecular diagnosis of visceral leishmaniasis in France: comparison between of a reference PCR method targeting kinetoplast DNA and a commercial kit targeting ribosomal DNA."

We modified the title into "Molecular diagnosis of visceral leishmaniasis a French study comparing a reference PCR method targeting kinetoplast DNA and a commercial kit targeting ribosomal DNA."

Line 15: "For direct diagnosis, quantitative PCR in blood is currently the most commonly used technique." Please review this statement, according to the literature, PCR in blood is not the most common for the diagnosis of VL.

We modified into "molecular diagnosis on blood and bone marrow are increasingly used techniques"

Line 19: Add: in France. The reference method, the authors stated, is not the reference method in everywhere.

We modified the sentence into "The "quany Leishmaniae, Clonit" kit was compared to a laboratory-developed method widely used in France and considered here as a reference method."

Line 29: "Leishmaniasis are protean diseases..."Please clarify this sentence.

We modified the sentence into " Leishmaniasis are protean a group of diseases"

Line 58: The copy number of the 18S rDNA gene is estimated to be about 20-40 (10)." According to the reference 8, the number of copies is around 200.

Reference 8 (Gow et al. 2022), in Table 3, gives 200 as the approximate copy number of 18S RNA, referring to a previous publication by the same group (Gow et al. 2019), in which they stated "The gene exists in between 50–200 copies per Leishmania genome, making it an excellent choice for a

pan-Leishmania detection assay” citing Srivastava et al. 2011. In this article, this range is indeed quoted, but without experimental data or other citations to back it up. In this revised version of our manuscript, we have preferred to retain the more consensual estimate of 20-40.

Line 151-162: "The rationale for this study is that more and more commercial kits for the molecular diagnosis of VL are available. The use of commercial kits should be encouraged, especially in the context of good laboratory practice and diagnostic accreditation. Nevertheless, it is fundamental that the performance of commercial kits be analyzed by experts independent of the manufacturer. This study was performed in the Parasitology-Mycology laboratory of the Montpellier University Hospital which hosts the French National Reference Center for leishmaniasis. Approximately 1,000 clinical samples are referred to and analyzed in the laboratory for VL and CL diagnosis per year. The study was divided into three parts. The first part used DNA standards in a one-tenth serial dilution assay to compare PCR efficiencies and to approach the sensitivity limits of the two methods. The second part used external quality controls (EQCs) to assess the performances of both qPCR technics on inter-laboratory comparison samples. The last part used patient samples to evaluate the performance and the agreement of the "quany Leishmaniae, Clonit" and the reference methods on clinical samples."

This paragraph could go on Discussion section.

Done

Line 163: Delete: "Better". The reader should be able to evaluate the data and to reach the similar conclusion.

Done

Add Probit analysis to compare the limit of detection between both tests.

We agree with this reviewer that Probit regression analysis is the method of choice for comparing detection limits. Data from Probit analysis is given for the Clonit kit in the manufacturer instruction sheet. In our study we particularly wanted to show that the Clonit method is not suitable for clinical practice, so we focused on comparing clinical samples. In the manuscript we have deleted (L157.184 and 263) references to comparisons of detection limits, which were insufficiently scientifically founded and diluted our message.

Line 179: Delete: "Better"

Done

Consider the following references:

León CM, Muñoz M, Hernández C, Ayala MS, Flórez C, Teherán A, Cubides JR, Ramírez JD. 2017. Analytical Performance of Four Polymerase Chain Reaction (PCR) and Real Time PCR (qPCR) Assays for the Detection of Six Leishmania Species DNA in Colombia. *Front Microbiol* 8:1907. <https://doi.org/10.3389/fmicb.2017.01907>

van Eys GJ, Schoone GJ, Ligthart GS, Alvar J, Evans DA, Terpstra WJ. Identification of 'Old World' Leishmania by DNA recombinant probes. *Mol Biochem Parasitol*. 1989 Apr;34(1):53-62. doi: 10.1016/0166-6851(89)90019-4

van Eys GJ, Schoone GJ, Kroon NC, Ebeling SB. 1992. Sequence analysis of small subunit ribosomal RNA genes and its use for detection and identification of Leishmania parasites. *Mol Biochem Parasitol* 51:133-142. [https://doi.org/10.1016/0166-6851\(92\)90208-2](https://doi.org/10.1016/0166-6851(92)90208-2)

We re-read these articles we already knew. We have quoted and added León et al. 2017 to the bibliographic list. We have not cited the other two, because, despite their qualities, they seemed to us to be outside the scope of this study, which focuses on visceral leishmaniasis.

Reviewer #2 (Comments for the Author):

Major comments

The study presented by Pasquier and his colleagues aims to compare a commercial visceral leishmaniasis diagnostic kit with a non-marketed but widely used reference method. Results show that the commercial kit does not reach the performance of the reference method, giving rise to several false negative results. The reviewer partially subscribes to the conclusions of the authors, nevertheless, he finds that their demonstration deserves to be better supported. This will be the meaning of the comments presented below. The main problem is the low number of samples tested in the successive phases of the study. The study relies on 5 external quality control and 19 clinical samples.

We now increased the total number of included samples to 40, with 35 clinical samples corresponding to 29 patients.

They have no discordant results on the 5 quality controls (4 positive and 1 negative) and only 1 discordant result on the 15 samples tested from their routine practice (sample 6, patient 1). Sample 12 from patient 6 is not really a discordant one since the authors identified the presence of an inhibitor in this sample and could solve the problem after dilution.

Finally, the main differences between the results obtained with the two methods is highlighted on a complementary set of four samples. Notably, these samples were selected because they had weak positive results with the reference method. Pasquier et al further show that the commercial technique failed to detect Leishmania DNA in three of these four samples.

From a methodological point of view, this way to select samples induces a bias. These samples are neither representative of the positive samples (since they are selected to be at the detection limit of the reference method), nor representative of all the samples tested in routine practice (which are mostly negative samples).

We agree with this reviewer. We have now included more samples in the study and the four additional samples from the previous version are now included in the study period, eliminating any possible bias. The 35 clinical samples, correspond to the 27 positive samples corresponding to the 23 consecutive patients analyzed for VL at Montpellier University Hospital between April 2019 and March 2022 and eight control samples/patients from the same period of time.

As a result, it is impossible to perform a sensitivity and specificity calculation. Furthermore, there is no estimate of the performance of the reference method in terms of sensitivity and specificity. Here, the specificity seems to be 100%, but the number of negative patients tested (5) is insufficient. As for the sensitivity, it cannot be calculated since the positivity of the PCR was one of the criteria for including four out of the 14 positive patients tested.

Nevertheless, the hypothesis formulated by Pasquier and his co-authors seems acceptable to me and I recommend continuing the study by including a greater number of routine samples in the comparison. Doing so they will probably obtain significant differences when estimating the performance of both methods and provide more convincing results.

We followed the reviewer's advice and included more samples. The clinical cohort now comprises 35 samples corresponding to 29 patients. We obtained 6 false negative results out of 27 positive

samples (22%). We hope to have convinced this reviewer that our hypothesis is supported by the data reported. It should be borne in mind that the disease is fairly rare in France.

Other comments

The abstract does not present any numerical value, only sentences indicating that the reference method gives better results.

Done, there are numerical values in the abstract of the new version of the manuscript.

The results section begins with two paragraphs which relate either to the description of the objectives of the study and its rationale (first paragraph), or to the description of the method and design of the study. It would be necessary to integrate them into the corresponding sections of the article.

Done, as also asked by reviewer 1, we integrated the first paragraph corresponding to the objectives and the rationale of the study to the Discussion section. Regarding the study design we integrated it at the end of Introduction.

We hope that you will now find our article suitable for publication in Microbiology Spectrum.

Yours faithfully,

Professeur Yvon STERKERS,

August 23, 2023

Dr. Yvon Sterkers
Universite de Montpellier
Montpellier cedex 5
France

Re: Spectrum02154-23R1 (Molecular diagnosis of visceral leishmaniasis a French study comparing a reference PCR method targeting kinetoplast DNA and a commercial kit targeting ribosomal DNA.)

Dear Dr. Yvon Sterkers:

Your manuscript has been accepted, and I am forwarding it to the ASM Journals Department for publication. You will be notified when your proofs are ready to be viewed.

Sincerely,

Denis Sereno
Editor, Microbiology Spectrum
